# Using genetic variants to evaluate the causal effect of cholesterol lowering on head and neck cancer risk: A Mendelian randomization study

Mark Gormley[1,2,3]*, James Yarmolinsky[1,3◉], Tom Dudding[1,2,3◉], Kimberley Burrows[1,3], Richard M. Martin[1,3,4], Steven Thomas[2,4], Jessica Tyrrell[5], Paul Brennan[6], Miranda Pring[2], Stefania Boccia[7,8], Andrew F. Olshan[9], Brenda Diergaarde[10], Rayjean J. Hung[11,12], Geoffrey Liu[12,13], Danny Legge[14], Eloiza H. Tajara[15], Patricia Severino[16], Martin Lacko[17], Andrew R. Ness[4], George Davey Smith[1,3], Emma E. Vincent[1,3,14◉], Rebecca C. Richmond[1,3◉]

1 MRC Integrative Epidemiology Unit, Population Health Sciences, Bristol Medical School, University of Bristol, Bristol, United Kingdom, 2 Bristol Dental Hospital and School, University of Bristol, Bristol, United Kingdom, 3 Department of Population Health Sciences, Bristol Medical School, University of Bristol, Bristol, United Kingdom, 4 National Institute for Health Research Bristol Biomedical Research Centre at the University Hospitals Bristol and Weston NHS Foundation Trust, University of Bristol, Bristol, United Kingdom, 5 University of Exeter Medical School, RILD Building, RD&E Hospital, Exeter, United Kingdom, 6 Genetic Epidemiology Group, World Health Organization, International Agency for Research on Cancer, Lyon, France, 7 Section of Hygiene, University Department of Life Sciences and Public Health, Università Cattolica del Sacro Cuore, Roma, Italia, 8 Department of Woman and Child Health and Public Health, Public Health Area, Fondazione Policlinico Universitario A. Gemelli IRCCS, Roma, Italy, 9 Department of Epidemiology, Gillings School of Global Public Health, University of North Carolina, Chapel Hill, North Carolina, United States of America, 10 Department of Human Genetics, Graduate School of Public Health, University of Pittsburgh, and UPMC Hillman Cancer Center, Pittsburgh, Pennsylvania, United States of America, 11 Prosserman Centre for Population Health Research, Lunenfeld-Tanenbaum Research Institute, Sinai Health System, Toronto, Canada, 12 Dalla Lana School of Public Health, University of Toronto, Toronto, Canada, 13 Princess Margaret Cancer Centre, Toronto, Canada, 14 School of Cellular and Molecular Medicine, University of Bristol, Bristol, United Kingdom, 15 School of Medicine of São José do Rio Preto, São Paulo, Brazil, 16 Albert Einstein Research and Education Institute, Hospital Israelita Albert Einstein, São Paulo, Brazil, 17 Department of Otorhinolaryngology and Head and Neck Surgery, Research Institute GROW, Maastricht University Medical Center, Maastricht, The Netherlands

◉ These authors contributed equally to this work.
* mark.gormley@bristol.ac.uk

**Data Availability Statement:** Summary-level analysis was conducted using publicly available GWAS data. Full summary statistics for the GAME-

## Abstract

Head and neck squamous cell carcinoma (HNSCC), which includes cancers of the oral cavity and oropharynx, is a cause of substantial global morbidity and mortality. Strategies to reduce disease burden include discovery of novel therapies and repurposing of existing drugs. Statins are commonly prescribed for lowering circulating cholesterol by inhibiting HMG-CoA reductase (HMGCR). Results from some observational studies suggest that statin use may reduce HNSCC risk. We appraised the relationship of genetically-proxied cholesterol-lowering drug targets and other circulating lipid traits with oral (OC) and oropharyngeal (OPC) cancer risk using two-sample Mendelian randomization (MR). For the primary analysis, germline genetic variants in *HMGCR*, *NPC1L1*, *CETP*, *PCSK9* and *LDLR* were used to proxy the effect of low-density lipoprotein cholesterol (LDL-C) lowering

ON GWAS can be accessed via dbGAP (OncoArray: Oral and Pharynx Cancer; study accession number: phs001202.v1.p1) and via the IEU OpenGWAS project https://gwas.mrcieu.ac.uk/. Access to UK Biobank (https://www.ukbiobank.ac.uk/) data is available to researchers through application. For the purpose of open access, the author has applied a CC BY public copyright licence to any Author Accepted Manuscript version arising from this submission. UK Biobank approval was given for this project (ID 40644 "Investigating aetiology, associations and causality in diseases of the head and neck") and UK Biobank GWAS data was also accessed under the application (ID 15825 "MR-Base: an online resource for Mendelian randomization using summary data"- Dr Philip Haycock." Code availability statement Two-sample MR analyses were conducted using the "TwoSampleMR" package in R (version 3.5.3). A copy of the code and all files used in this analysis is available at: https://github.com/rcrichmond/cholesterol_lowering_headandneckcancer.

**Funding:** M.G. was a National Institute for Health Research (NIHR) academic clinical fellow and is currently supported by a Wellcome Trust GW4-Clinical Academic Training PhD Fellowship. This research was funded in part, by the Wellcome Trust [Grant number 220530/Z/20/Z]. For the purpose of open access, the author has applied a CC BY public copyright licence to any Author Accepted Manuscript version arising from this submission. UK Biobank approval was given for this project (ID 40644 "Investigating aetiology, associations and causality in diseases of the head and neck") and UK Biobank GWAS data was also accessed under the application (ID 15825 "MR-Base: an online resource for Mendelian randomization using summary data"- Dr Philip Haycock). R.C.R. is a de Pass VC research fellow at the University of Bristol. J.T. is supported by an Academy of Medical Sciences (AMS) Springboard award and Diabetes UK (SBF004\1079). R.M.M. was supported by a Cancer Research UK (C18281/A29019) programme grant (the Integrative Cancer Epidemiology Programme) and is part of the Medical Research Council Integrative Epidemiology Unit at the University of Bristol supported by the Medical Research Council (MC_UU_00011/1, MC_UU_00011/7) and the University of Bristol. JY is supported by a Cancer Research UK Population Research Postdoctoral Fellowship (C68933/A28534). R.M.M. and A.R.N. are supported by the National Institute for Health Research (NIHR) Bristol Biomedical Research Centre which is funded by the National Institute for Health Research (NIHR) and is a partnership between University Hospitals Bristol and Weston NHS

therapies. In secondary analyses, variants were used to proxy circulating levels of other lipid traits in a genome-wide association study (GWAS) meta-analysis of 188,578 individuals. Both primary and secondary analyses aimed to estimate the downstream causal effect of cholesterol lowering therapies on OC and OPC risk. The second sample for MR was taken from a GWAS of 6,034 OC and OPC cases and 6,585 controls (GAME-ON). Analyses were replicated in UK Biobank, using 839 OC and OPC cases and 372,016 controls and the results of the GAME-ON and UK Biobank analyses combined in a fixed-effects meta-analysis. We found limited evidence of a causal effect of genetically-proxied LDL-C lowering using HMGCR, NPC1L1, CETP or other circulating lipid traits on either OC or OPC risk. Genetically-proxied PCSK9 inhibition equivalent to a 1 mmol/L (38.7 mg/dL) reduction in LDL-C was associated with an increased risk of OC and OPC combined (OR 1.8 95%CI 1.2, 2.8, p = 9.31 x10$^{-05}$), with good concordance between GAME-ON and UK Biobank ($I^2$ = 22%). Effects for PCSK9 appeared stronger in relation to OPC (OR 2.6 95%CI 1.4, 4.9) than OC (OR 1.4 95%CI 0.8, 2.4). LDLR variants, resulting in genetically-proxied reduction in LDL-C equivalent to a 1 mmol/L (38.7 mg/dL), reduced the risk of OC and OPC combined (OR 0.7, 95%CI 0.5, 1.0, p = 0.006). A series of pleiotropy-robust and outlier detection methods showed that pleiotropy did not bias our findings. We found limited evidence for a role of cholesterol-lowering in OC and OPC risk, suggesting previous observational results may have been confounded. There was some evidence that genetically-proxied inhibition of PCSK9 increased risk, while lipid-lowering variants in LDLR, reduced risk of combined OC and OPC. This result suggests that the mechanisms of action of PCSK9 on OC and OPC risk may be independent of its cholesterol lowering effects; however, this was not supported uniformly across all sensitivity analyses and further replication of this finding is required.

## Author summary

Author summary

This study aimed to determine if genetically-proxied cholesterol-lowering drugs (such as statins which target HMGCR) and genetically-proxied circulating lipid traits (e.g., low-density lipoprotein cholesterol) have a causal effect on oral and oropharyngeal cancer risk. There was little evidence that genetically-proxied inhibition of HMGCR (target of statins), NPC1L1 (target of ezetimibe) and CETP (target of CETP inhibitors) influences oral or oropharyngeal cancer risk. Similarly, there was little evidence of an effect of circulating lipid traits on oral or oropharyngeal cancer risk. We did find some evidence that genetically-proxied inhibition of PCSK9 increases, while lipid-lowering variants in LDLR reduce oral and oropharyngeal cancer risk. Our findings suggest that the results of previous observational studies examining the effect of statins on oral and oropharyngeal risk may have been confounded. The mechanism of action of PCSK9 may be independent of cholesterol-lowering, however further replication of this finding in other head and neck cancer datasets is required.

## Introduction

Head and neck squamous cell carcinoma (HNSCC), which includes cancers of the oral cavity (OC) and oropharynx (OPC), is the sixth most common cancer in the world, with over

Foundation Trust and the University of Bristol. This publication presents data from the Head and Neck 5000 study. The study was a component of independent research funded by the NIHR under its Programme Grants for Applied Research scheme (RP-PG-0707-10034). The views expressed in this publication are those of the author(s) and not necessarily those of the NHS, the NIHR or the Department of Health. Core funding was also provided through awards from Above and Beyond, University Hospitals Bristol Research Capability Funding and the NIHR Senior Investigator award to A.R.N. Human papillomavirus (HPV) serology was supported by a Cancer Research UK Programme Grant, the Integrative Cancer Epidemiology Programme (grant number: C18281/A19169). B.D. and the University of Pittsburgh head and neck cancer case-control study are supported by US National Institutes of Health (NIH) grants: P50 CA097190, P30 CA047904 and R01 DE025712. The genotyping of the HNSCC cases and controls was performed at the Center for Inherited Disease Research (CIDR) and funded by the US National Institute of Dental and Craniofacial Research (NIDCR; 1X01HG007780-0). A.F.O. and the University of North Carolina (UNC) CHANCE study was supported in part by the Division of Cancer Prevention, National Cancer Institute (R01-CA90731). E.E.V and D.L are supported by Diabetes UK (17/0005587). E.E.V is also supported by the World Cancer Research Fund (WCRF UK), as part of the World Cancer Research Fund International grant programme (IIG_2019_2009). E.H.T and P.S. were supported by the GENCAPO/FAPESP grant (10/51168-0). The funders had no role in study design, data collection and analysis, decision to publish, or preparation of the manuscript.

**Competing interests:** The authors have declared that no competing interests exist.

550,000 new cases and 300,000 deaths annually [1, 2]. Despite some modest improvements in the treatment of HNSCC, survival ranges between 19–59% at 10 years [3] and recurrence rates remain high [4]. Patients often undergo a combination of surgery, radiotherapy and chemotherapy which can result in significant morbidity [5]. Established risk factors include smoking, alcohol and human papilloma virus (HPV), the latter mainly linked with oropharyngeal cancer [6, 7]. Given that in contemporary cohorts, around 70% of OPC cases (versus <5% of OC cases) are HPV driven and often present in younger populations, oral and oropharyngeal tumours are considered distinct disease entities, with different risk factor profiles [8]. Despite smoking cessation, alcohol reduction measures and the implementation of HPV vaccination in some areas, HNSCC remains a major global health problem [2]. Novel strategies for prevention of HNSCC are required, in particular for those at high risk and one approach is to identify novel risk factors which can be easily modified, for example by repurposing existing drugs [9].

Statins are one of the most commonly prescribed medications worldwide. They are prescribed to reduce levels of circulating total and low-density lipoprotein cholesterol (LDL-C), with proven preventative and therapeutic effects in cardiovascular disease and a good safety profile [10, 11]. Statins inhibit 3-hydroxy-3-methylglutaryl-coenzyme A reductase (HMG-CoA reductase (HMGCR)), the rate-limiting enzyme in the mevalonate pathway, responsible for cholesterol and steroid hormone synthesis [12]. Other clinically approved drugs that target cholesterol metabolism via different mechanisms include ezetimibe (targeting Niemann-Pick C1-like protein (NPC1L1)) and proprotein convertase subtilisin/kexin type 9 (PCSK9) inhibitors such as evolocumab or alirocumab. These agents act by reducing the intestinal absorption of cholesterol or by enhancing uptake of LDL-C through increased cellular membrane expression of the LDL-receptor (LDLR), respectively. Conversely, cholesteryl ester transfer protein inhibitors (CETP) substantially increase levels of high-density lipoprotein cholesterol (HDL-C), lower levels of LDL-C and enhance reverse cholesterol transport [13].

Cholesterol is vital for a variety of key cellular functions, including membrane integrity, signalling, protein synthesis and cell cycle progression. Therefore, modulation of cholesterol synthesis has the potential to influence several hallmarks of tumourigenesis including cell migration and proliferation [14]. In an experimental study, mice given oral daily doses of simvastatin two weeks prior to subcutaneous injection of FaDu (HPV-negative hypopharyngeal tumour cells), showed a significant reduction in tumour growth. This study was designed to mimic a clinical scenario where patients who present with a tumour may have been taking the medication prior to tumour initiation [15], and suggested that simvastatin antagonises tumour metabolic reprogramming, another important hallmark of cancer [15]. Mechanistic support for the role of LDL-C lowering in cancer development comes largely from the fact that lipids are themselves major cell membrane components essential for cell division and maintaining tissue integrity. Changes in lipid levels have been reported associated with tumour development [16, 17].

However, the evidence that cholesterol-lowering drugs may reduce the risk of head and neck cancer *in vivo* is limited. Some observational studies report an inverse association of taking statins with both head and neck cancer risk [18] and cancer survival [14], but others indicate little evidence of any effect on cancer [19]. Observational studies are not randomised and are susceptible to reverse causality and/or confounding [19]. Mendelian randomization (MR) is an approach that uses germline genetic proxies (referred to as instruments) to help appraise causal effects of potentially modifiable extrinsic exposures or intrinsic traits with disease [20–22]. It has also been used to estimate therapeutic potential by investigating genetic variation at drug targets [23]. Numerous single nucleotide polymorphisms (SNPs) are associated with lower levels of circulating LDL-C [24, 25] and inheriting an LDL-C lowering allele has been proposed to be analogous to being assigned life-long treatment with a cholesterol lowering drug [26]. In this way, germline genetic variants may serve as proxies for exposure to potential

pharmacological agents which are less likely than observational measures to be subject to reverse causation or confounding. Genetic proxies can therefore be used to predict both the likely beneficial and adverse effects of long-term modulation of the drug targets on disease.

MR has previously demonstrated the protective effect of cholesterol-lowering drugs on cardiovascular disease risk [27], but also that inhibition of HMGCR and PCSK9 may have an adverse effect on diabetes risk [28]. Of relevance to cancer, some recent MR studies have shown that genetically-proxied inhibition of HMGCR may be protective against overall cancer [29] and epithelial ovarian cancer [30] risk. Our aim was to use MR to appraise the causal nature and mechanistic basis of the relationship between cholesterol-lowering and risk of oral and oropharyngeal cancer by investigating germline variation in HMGCR, NPC1L1, CETP, PCSK9 and LDLR, and other related lipid traits such as circulating LDL-C.

## Methods

### Identifying cholesterol-lowering genetic instruments

For the primary analysis, SNPs in *HMGCR*, *NPC1L1*, *CETP*, *PCSK9* and *LDLR* were used to proxy the effect of lipid-lowering therapies and to estimate the downstream effect of manipulating these targets on OC and OPC risk. SNPs were identified within 100 kb on either side of the target gene (*HMGCR*, *NPC1L1*, *CETP*, *PCSK9* and *LDLR)* that were associated with LDL-C levels. Variants were robustly associated with LDL-C in a meta-analysis of genome-wide association studies (GWAS) involving 188,578 individuals primarily (96%) of European ancestry in the Global Lipids Genetic Consortium (GLGC) [25]. As previously described by Ference et al., SNPs were iteratively selected for inclusion in order of decreasing magnitude of association (effect size) with LDL-C. All polymorphisms had a *p*-value for association with LDL-C of $<5$ x$10^{-8}$ and low linkage disequilibrium (LD) (defined as $r^2 <0.2$) with all other SNPs that were included in the score [28]. Multiple papers have used these genetic instruments at this threshold, to demonstrate causal effects in cardiovascular disease [13, 27, 31], diabetes [28] and ovarian cancer [30]. In secondary analyses, SNPs were used to proxy circulating levels of LDL-C, HDL-C, total triglyceride, total cholesterol, apolipoprotein A and B. Betas represented the change in lipid trait levels per copy of the effect allele. SNPs utilised in the secondary analysis SNPs were already independently ($r^2 <0.001$) associated with the respective traits in large GWAS which have been described previously [25, 32].

### Summary level genetic data on oral and oropharyngeal cancer from GAME-ON

We estimated the effects of the cholesterol-lowering genetic variants on risk of OC and OPC using GWAS performed on 6,034 cases and 6,585 controls from 12 studies which were part of the Genetic Associations and Mechanisms in Oncology (GAME-ON) Network [33]. The study population included participants from Europe (45.3%), North America (43.9%) and South America (10.8%). Cancer cases comprised the following the International Classification of Diseases (ICD) codes: oral cavity (C02.0-C02.9, C03.0-C03.9, C04.0-C04.9, C05.0-C06.9) oropharynx (C01.9, C02.4, C09.0-C10.9), hypopharynx (C13.0-C13.9), overlapping (C14 and combination of other sites) and 25 cases with unknown ICD code (other). A total of 954 individuals with cancers of hypopharynx, unknown code or overlapping cancers were excluded. Genomic DNA isolated from blood or buccal cells was genotyped at the Center for Inherited Disease Research (CIDR) using an Illumina OncoArray, custom designed for cancer studies by the OncoArray Consortium [34]. In GAME-ON, all SNPs with a call rate of <95% were excluded. Given the ethnic heterogeneity of the study population, the dataset was divided by

geographical region and SNPs within each region that showed deviation from Hardy-Weinberg Equilibrium (HWE) in controls ($p = <1 \times 10^{-7}$) were excluded. Principal component analysis (PCA) was performed using approximately 10,000 common markers in low LD ($r^2 < 0.004$), minor allele frequency (MAF) >0.05 and 139 population outliers were removed. Full details of the included studies, as well as the genotyping and imputation performed, have been described previously [33, 35].

### Two-sample Mendelian randomization

Two-sample MR was conducted using the "TwoSampleMR" package in R (version 3.5.3), by integrating SNP associations for cholesterol-lowering (sample 1) with those for OC and OPC in GAME-ON (sample 2). For those SNPs instrumenting LDL-C lowering, we first extracted summary statistics for the associations with OC and OPC from GAME-ON. We next performed harmonisation of the direction of effects between the cholesterol-lowering exposures and outcome (OC or OPC) where, for each variant, the allele designated the 'exposure allele' was associated with lower LDL-C levels and palindromic SNPs were aligned when MAFs were <0.3 or were otherwise excluded. In our primary analysis, four palindromic SNPs, one in *HMGCR* (rs2006760), the other in *PCSK9* (rs2149041) and two in *CETP* (rs5880, rs9929488) were removed. In the secondary analysis with other lipid traits, 11 palindromic SNPs (rs1936800, rs2288912, rs7112577, rs964184, rs150617279, rs1883711, rs4722043, rs2156552, rs2954029, rs581080, rs7534572) were removed.

Individual effect-estimates for each SNP were calculated using the Wald ratio, by dividing the SNP-outcome association by the SNP-exposure association. Multiple SNPs were then combined into multi-allelic instruments using the random-effects inverse-variance weighted (IVW) meta-analysis method, for each of the genes *HMGCR*, *NPC1L1*, *CETP*, *PCSK9* and *LDLR*. This meta-analysis was undertaken to increase the proportion of variance in drug targets and LDL-C lowering explained by each instrument, and thus improve statistical power and the precision of our estimates [36]. The analysis produced an estimate of the effect of the risk factor on OC and OPC risk. As the instruments for *HMGCR*, *NPC1L1*, *CETP*, *PCSK9* and *LDLR* were in weak LD ($r^2 < 0.2$), we accounted for this correlation between SNPs in the primary analysis using LDlink (4.0 Release) which employs Phase 3 (Version 5) of the 1000 Genomes Project and variant rs numbers based on dbSNP [37]. Correlation matrices were inserted as an MRInput object, resulting in MR methods which altered the weightings for correlated SNPs [38–40]. For circulating lipid traits, a more stringent $r^2 < 0.001$ was already applied in the initial GWAS [25], so we did not account further for correlation in the secondary analysis. We computed odds ratios (OR) which represent the change in odds of oral and oropharyngeal squamous cell carcinoma per genetically-proxied inhibition of the drug target, equivalent to a 1 mg/dl decrease in LDL-C. The OR was scaled to be per 1 mmol/L decrease by dividing the LDL-C lowering effect (beta) and standard error (se) measured in mg/dL by 38.7 [30]. The betas and standard errors for *HMGCR*, *NPC1L1*, *CETP*, *PCSK9* and *LDLR* SNPs were also converted to reflect the cholesterol-lowering effect, given the "TwoSampleMR" package preference to automatically change these into a positive direction of effect (i.e., lipid increasing). The correct direction of effect was checked using a positive control of coronary heart disease from the CARDIoGRAM GWAS data [41]. MR was also used to examine the effect of circulating levels of LDL-C, HDL-C, total triglyceride, total cholesterol, apolipoprotein A and B levels directly with OC and OPC cancer risk. The ORs in this analysis represent the change in odds of oral or oropharyngeal squamous cell carcinoma, per SD unit increase in lipid trait.

The IVW method can provide an unbiased effect estimate in the absence of horizontal pleiotropy or when horizontal pleiotropy is balanced [42]. We therefore performed sensitivity analyses to evaluate the potential for unbalanced horizontal pleiotropy, where genetic variants influence

two or more traits through independent biological pathways. To ensure the genetic instrument was associated with the instrument it was proxying, estimates of the proportion of variance in each risk factor explained by the instrument ($R^2$) and F-statistics were generated. An F-statistic of <10 is indicative of a weak instrument which may be subject to weak instrument bias. To account for directional pleiotropy, we compared the IVW results with three MR sensitivity analyses, which each make different assumptions: MR Egger [43], weighted median [44] and weighted mode [45]. While these three methods are best used when genetic instruments consist of a large numbers of independent SNPs, since $r^2$ values between SNPs in our instruments were low ($r^2$ <0.2), we a-priori decided to include them, with a further sensitivity analysis to account for correlation in the MR Egger analysis. The weighted median stipulates that at least 50% of the weight in the analysis stems from variants that are valid instruments [44], while the weighted mode requires that the largest subset of instruments which identify the same causal effect to be valid instruments [45]. MR-Egger can provide unbiased estimates even when all SNPs in an instrument violate the exclusion restriction assumption (i.e., affect the outcome by means other than via the risk factor of interest). However, there must be negligible measurement error (NOME) [46] in the genetic instrument and the InSIDE (Instrument Strength Independent of Direct Effect) assumption must be satisfied [43]. Where there was evidence of violation of the NOME assumption, this was assessed using the $I^2$ statistic and MR-Egger was performed with simulation extrapolation (SIMEX) correction [46]. To further assess the robustness of findings, we examined evidence of heterogeneity in the individual SNP estimates using the Cochran Q-statistic, which may indicate the presence of invalid instruments (e.g., due to horizontal pleiotropy) [47]. Scatter and leave-one-out plots were produced to evaluate influential outliers and MR-PRESSO (Mendelian Randomization Pleiotropy RESidual Sum and Outlier) was used to detect and correct for potential outliers (where Q-statistic p <0.05) [48]. For any positive findings, we ran colocalisation analysis using the 'coloc' package in R [49], to test if there was violation of the MR exclusion restriction assumption. This can be generated through instrument-exposure and instrument-outcome associations, driven by distinct causal variants that are in LD with each other. The 'coloc' package in R enumerates every possible configuration of causal variants for two traits, calculating the support for that causal model in the form of a Bayes factor, assuming that at most one causal variant per trait exists in the region and there are similar LD structures across the two samples. Approximate Bayes factor colocalisation analysis fine maps each trait under a single causal variant assumption and then integrates these over two posterior distributions to estimate probabilities that those variants are shared [49]. A posterior probability of $\geq 0.80$ is considered evidence to support a particular configuration tested in 'coloc'. Using genomic regions of 1 Mb either side of the lead variant for the genes of interest, we investigated whether findings reflect shared causal variants.

## Stratification by cancer subsite

Given the difference in established aetiology (i.e. smoking, alcohol and HPV) at each HNSCC subsite [6], we performed MR analyses with stratification by cancer subsite to evaluate potential heterogeneity in effects. For this, we used GWAS summary data on a subset of 2,641 OPC cases and 2,990 OC cases from the 6,034 HNSCC cases and the 6,585 common controls in the GAME-ON GWAS [33].

## Replication in UK Biobank

UK Biobank data was used as a replication dataset for primary analyses. A GWAS was performed on 839 combined OC and OPC cases and 372,016 controls, with two further stratified GWAS for OC (n = 357) and OPC (n = 494). UK Biobank is a large population-based cohort study that recruited over 500,000 men and women aged between 37 and 73 years between

2006 and 2010 throughout the UK. It received ethical approval from the National Health Service North West Centre for Research Ethics Committee (reference: 11/NW/0382). Details of genotyping quality control, phasing and imputation are described elsewhere [50]. Participant records are linked to cancer registry data and HNSCC was grouped using the same ICD-codes as described above. Squamous cell carcinoma cases were identified using histology codes 8070–8078. UK Biobank GWAS analyses were adjusted for sex and genotyping array and performed in BOLT-LMM, a mixed model that accounts for population stratification and relatedness [51, 52]. Primary MR analyses as described above in GAME-ON were repeated in UK Biobank data.

## Meta-analysis of results

We performed both fixed-effects and random-effects meta-analysis of the MR estimates in GAME-ON and UK Biobank using the R package 'meta'. However, we focus more on the fixed-effects estimates since we assume that the causal effect is constant between the studies [53]. Heterogeneity between study populations was assessed using $I^2$ statistic [54].

## Results

### Primary analysis in GAME-ON

In total, 5 SNPs in *HMGCR* (rs12916, rs17238484, rs5909, rs2303152, rs10066707) were used to proxy HMG-CoA reductase inhibition (statins); 5 SNPs in *NPC1L1* (rs217386, rs2073547, rs7791240, rs10234070, rs2300414) proxied NPC1L1 inhibition (ezetimibe); 6 SNPs in *CETP* (rs9989419, rs12708967, rs3764261, rs1800775, rs1864163, rs289714) proxied CETP inhibition; 6 SNPs in *PCSK9* (rs11206510, rs2479409, rs2479394, rs10888897, rs7552841, rs562556) proxied PCSK9 inhibition; 3 SNPs proxied cholesterol-lowering in *LDLR* (rs6511720, rs1122608, rs688) (LDL-receptor inhibition). Further details of these SNP effects are given in **Table 1**.

There was limited evidence of an effect of genetically-proxied inhibition of HMGCR and NPC1L1 on combined OC and OPC risk (OR IVW 1.1; 95%CI 0.6, 1.9, p = 0.82 and 1.0; 95% CI 0.4, 2.7, p = 0.99), respectively (**Table 2** and **S1 Fig**). A similar result was found for genetically-proxied LDL-C lowering inhibition of CETP on OC and OPC combined (OR IVW 1.3; 95%CI 0.6, 2.6, p = 0.49) (**Table 2** and **S1 Fig**). However, higher risk of combined OC and OPC was found in relation to genetically-proxied PCSK9 inhibition, equivalent to a 1 mmol/L (38.7 mg/dL) reduction in LDL-C (OR IVW 2.1; 95%CI 1.2, 3.4, p = 0.01; **Table 2** and **Fig 1**). This is in contrast to the reduction in odds seen in relation to cardiovascular disease using the same instrument (OR IVW 0.6; 95%CI 0.4, 0.8, p $<1$ x$10^{-03}$; **Fig 2**) in 60,801 cases and 123,504 control subjects enrolled in the CARDIoGRAM consortia studies [41]. There was also some evidence that LDLR variants, resulting in genetically-proxied reduction in LDL-C equivalent to a 1 mmol/L (38.7 mg/dL), reduced the risk of combined OC and OPC (OR IVW; 0.7; 95% CI 0.4, 1.0, p = 0.05; **Table 2** and **Fig 1**). The combined OC/OPC results were robust to multiple testing in the main analysis conducted (i.e., IVW for the drug targets = $p <0.05/5 = 0.001$).

### Stratification by cancer subsite in GAME-ON

When stratified by subsite, the adverse effect of PCSK9 appeared to be mainly in the oropharynx, (OR IVW 3.5; 95%CI 1.6, 7.7, p = 2 x$10^{-03}$), with limited evidence in the oral cavity (OR IVW 1.6; 0.9, 2.94, p = 0.11) (**Table 2** and **Fig 1**). The effects appeared stronger in OPC versus OC, but with overlapping confidence intervals. LDLR was associated with a reduction in risk of OC (OR IVW 0.5; 95%CI 0.3, 0.9, p = 0.01), but there was little evidence of an association with OPC (OR IVW 0.9; 95%CI 0.5, 1.5, p = 0.69) (**Table 2** and **Fig 1**). For both PCSK9 and

**Table 1. Detailed summary of LDL-C lowering genetic variants in *HMGCR*, *NPC1L1*, *CETP*, *PCSK9* and *LDLR* variants from the in Global Lipids Genetics Consortium (GLGC).**

| Target | SNP | Pathway | EA | OA | EAF | Beta | se | *P*-value |
|---|---|---|---|---|---|---|---|---|
| *HMGCR* | rs12916 | LDL-C | T | C | 0.57 | -0.06061 | 0.003 | 7.79E-78 |
| | rs17238484 | LDL-C | G | T | 0.75 | -0.05184 | 0.005 | 1.35E-21 |
| | rs5909 | LDL-C | G | A | 0.90 | -0.05102 | 0.007 | 4.93E-13 |
| | rs2303152 | LDL-C | G | A | 0.88 | -0.03498 | 0.005 | 1.04E-09 |
| | rs10066707 | LDL-C | G | A | 0.58 | -0.0411 | 0.005 | 2.97E-19 |
| | rs2006760* | LDL-C | C | G | 0.81 | -0.04407 | 0.006 | 1.67E-13 |
| *NPC1L1* | rs217386 | LDL-C | A | G | 0.41 | -0.02908 | 0.003 | 1.20E-19 |
| | rs2073547 | LDL-C | A | G | 0.81 | -0.03885 | 0.004 | 1.92E-21 |
| | rs7791240 | LDL-C | T | C | 0.91 | -0.03404 | 0.005 | 1.84E-10 |
| | rs10234070 | LDL-C | C | T | 0.90 | -0.02363 | 0.005 | 1.52E-06 |
| | rs2300414 | LDL-C | G | A | 0.93 | -0.02828 | 0.006 | 5.45E-06 |
| *CETP* | rs3764261 | LDL-C | A | C | 0.29 | -0.04471 | 0.004 | 2.22E-34 |
| | rs1800775 | LDL-C | A | C | 0.48 | -0.03487 | 0.003 | 8.54E-24 |
| | rs1864163 | LDL-C | G | A | 0.73 | -0.03698 | 0.004 | 7.97E-21 |
| | rs9929488* | LDL-C | G | C | 0.70 | -0.03159 | 0.004 | 8.15E-13 |
| | rs9989419 | LDL-C | G | A | 0.60 | -0.02344 | 0.004 | 2.49E-12 |
| | rs12708967 | LDL-C | T | C | 0.80 | -0.02963 | 0.004 | 3.47E-11 |
| | rs289714 | LDL-C | A | G | 0.79 | -0.03032 | 0.005 | 2.85E-10 |
| | rs5880* | LDL-C | G | C | 0.94 | -0.03979 | 0.008 | 1.59E-06 |
| *PCSK9* | rs11206510 | LDL-C | C | T | 0.15 | -0.06871 | 0.001 | 2.38E-53 |
| | rs2479409 | LDL-C | A | G | 0.67 | -0.05309 | 0.001 | 2.52E-50 |
| | rs2149041* | LDL-C | C | G | 0.84 | -0.05259 | 0.001 | 1.44E-35 |
| | rs2479394 | LDL-C | A | G | 0.72 | -0.03192 | 0.001 | 1.58E-19 |
| | rs10888897 | LDL-C | T | C | 0.40 | -0.04192 | 0.001 | 8.43E-31 |
| | rs7552841 | LDL-C | C | T | 0.63 | -0.04589 | 0.001 | 5.40E-15 |
| | rs562556 | LDL-C | G | A | 0.19 | -0.05292 | 0.002 | 6.16E-21 |
| *LDLR* | rs6511720 | LDL-C | T | G | 0.11 | -0.17482 | 0.004 | 3.69E-54 |
| | rs1122608 | LDL-C | T | G | 0.23 | -0.05473 | 0.003 | 2.02E-86 |
| | rs688 | LDL-C | C | T | 0.56 | -0.04465 | 0.003 | 3.04E-48 |

Abbreviations: EA, effect allele or low-density lipoprotein-cholesterol (LDL-C) lowering allele; OA = other or non-effect allele; EAF, effect allele frequency; se = standard error.

* Palindromic SNPs removed. Beta represents the change in LDL-C levels per copy of the effect allele. For SI conversion of mmol/L to mg/dL, multiply by 38.7.

LDLR associations, the direction of effect was generally consistent across the four MR methods tested (**Table 2**).

## Secondary analysis in GAME-ON

SNPs were also used to proxy circulating levels of LDL-C (77 SNPs), high-density lipoprotein cholesterol (HDL-C) (85 SNPs), total triglyceride (54 SNPs), total cholesterol (82 SNPs), apolipoprotein A (9 SNPs) and apolipoprotein B (14 SNPs) (**S1 Table**). There was limited evidence of an effect of any of these other lipid traits on either OC or OPC (**Table 3 and S2 Fig**).

## Sensitivity analyses

IVW, MR Egger, weighted median, simple and weighted mode were carried out, in addition to IVW analysis accounting for LD structure (**S2 Table**). The results adjusting for SNP

**Table 2. Mendelian randomization results of genetically-proxied inhibition of HMGCR, NPC1L1, CETP, PCSK9 and LDLR with risk of oral and oropharyngeal cancer including sensitivity analyses in GAME-ON.**

| | Outcome | Exposure/Outcome dataset | Outcome N | Number of SNPs | IVW OR (95% CI) | P | Weighted median OR (95% CI) | P | Weighted mode OR (95% CI) | P | MR-Egger OR (95%CI) | P |
|---|---|---|---|---|---|---|---|---|---|---|---|---|
| HMGCR | Oral/ Oropharyngeal cancer | GAME-ON/GLGC | 6,034 | 5 | 1.07 (0.62, 1.84) | 0.82 | 1.20 (0.63, 2.28) | 0.58 | 1.20 (0.57, 2.50) | 0.66 | 1.33 (0.07, 26.58) | 0.86 |
| | Oral cancer | GAME-ON/GLGC | 2,990 | 5 | 1.49 (0.75, 2.96) | 0.25 | 1.66 (0.73, 3.78) | 0.23 | 1.69 (0.65, 4.42) | 0.35 | 1.25 (0.03, 55.80) | 0.91 |
| | Oropharyngeal cancer | GAME-ON/GLGC | 2,641 | 5 | 0.90 (0.43, 1.85) | 0.77 | 1.01 (0.43, 2.34) | 0.99 | 1.08 (0.44, 2.64) | 0.88 | 0.77 (0.01, 45.07) | 0.91 |
| NPC1L1 | Oral/ Oropharyngeal cancer | GAME-ON/GLGC | 6,034 | 5 | 1.01 (0.38, 2.69) | 0.99 | 0.90 (0.29, 2.82) | 0.86 | 0.86 (0.24, 3.08) | 0.83 | 0.22 (0.00, 102.42) | 0.66 |
| | Oral cancer | GAME-ON/GLGC | 2,990 | 5 | 1.02 (0.30, 3.41) | 0.98 | 1.20 (0.29, 5.05) | 0.80 | 1.29 (0.22, 7.41) | 0.79 | 0.09 (0.00, 160.66) | 0.57 |
| | Oropharyngeal cancer | GAME-ON/GLGC | 2,641 | 5 | 0.60 (0.16, 2.25) | 0.45 | 0.60 (0.12, 3.04) | 0.53 | 0.60 (0.10, 3.73) | 0.62 | 0.20 (0.01, 688.64) | 0.72 |
| CETP | Oral/ Oropharyngeal cancer | GAME-ON/GLGC | 6,034 | 6 | 1.28 (0.64, 2.55) | 0.49 | 1.24 (0.54, 2.81) | 0.61 | 1.25 (0.43, 3.64) | 0.71 | 0.82 (0.02, 27.60) | 0.92 |
| | Oral cancer | GAME-ON/GLGC | 2,990 | 6 | 1.65 (0.70, 3.88) | 0.25 | 1.69 (0.61, 4.70) | 0.31 | 1.63 (0.43, 5.87) | 0.49 | 0.42 (0.01, 32.58) | 0.72 |
| | Oropharyngeal cancer | GAME-ON/GLGC | 2,641 | 6 | 1.12 (0.45, 2.77) | 0.81 | 1.05 (0.35, 3.20) | 0.93 | 0.73 (0.17, 3.07) | 0.69 | 1.50 (0.02, 142.22) | 0.87 |
| PCSK9 | Oral/ Oropharyngeal cancer | GAME-ON/GLGC | 6,034 | 6 | 2.05 (1.24, 3.38) | 0.01 | 2.21 (1.12, 4.08) | 0.01 | 2.19 (0.81, 5.92) | 0.18 | 1.83 (0.17, 20.00) | 0.65 |
| | Oral cancer | GAME-ON/GLGC | 2,990 | 6 | 1.62 (0.89, 2.94) | 0.11 | 1.80 (0.90, 3.62) | 0.10 | 2.00 (0.73, 5.48) | 0.23 | 2.17 (0.15, 32.13) | 0.60 |
| | Oropharyngeal cancer | GAME-ON/GLGC | 2,641 | 6 | 3.49 (1.58, 7.68) | 2.00E-03 | 3.19 (1.40, 7.26) | 0.01 | 2.73 (0.86, 8.70) | 0.15 | 1.99 (0.04, 92.11) | 0.74 |
| LDLR | Oral/ Oropharyngeal cancer | GAME-ON/GLGC | 6,034 | 3 | 0.65 (0.42, 1.00) | 0.05 | 0.71 (0.47, 1.08) | 0.11 | 0.74 (0.47, 1.15) | 0.31 | 0.90 (0.41, 1.95) | 0.83 |
| | Oral cancer | GAME-ON/GLGC | 2,990 | 3 | 0.53 (0.32, 0.87) | 0.01 | 0.56 (0.32, 0.98) | 0.04 | 0.57 (0.32, 1.04) | 0.21 | 0.56 (0.18, 1.76) | 0.50 |
| | Oropharyngeal cancer | GAME-ON/GLGC | 2,641 | 3 | 0.90 (0.53, 1.52) | 0.69 | 0.91 (0.55, 1.53) | 0.73 | 1.08 (0.62, 1.89) | 0.81 | 1.56 (0.64, 3.82) | 0.51 |

Abbreviations: IVW, inverse variance weighted; OR, odds ratio; CI, confidence intervals; P, *p*-value.

OR represents the exponential change in odds of oral/ oropharyngeal squamous cell carcinoma per genetically-proxied inhibition of drug target equivalent to a 1 mmol/L decrease in LDL-C.

correlation followed the same pattern as the main results (**S3 Table**). There was limited evidence of weak instrument bias being present (F-statistic >10) and the proportion of variance in the phenotype ($R^2$) explained by the genetic instruments ranged from 0.1 to 6% (**S4 Table**).

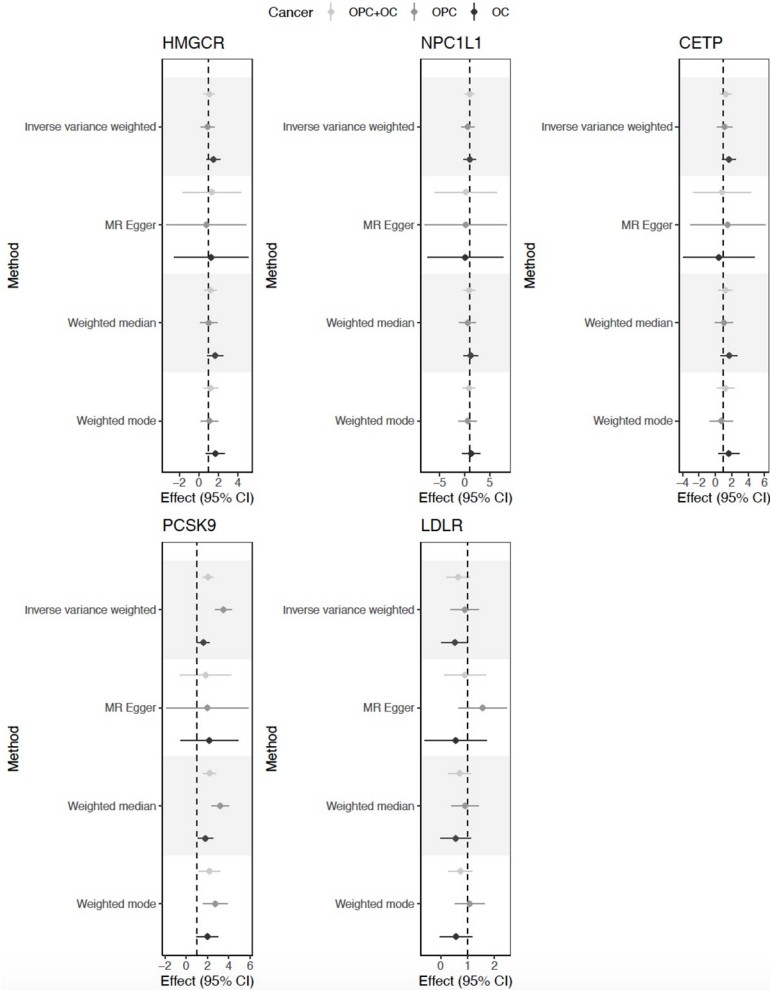

**Fig 1. Forest plot showing the causal effects of HMGCR, NPC1L1, CETP, PCSK9 and LDLR variants on the oral and oropharyngeal cancer subsites in GAME-ON.** Effect estimates on oral and oropharyngeal cancer are reported on the log odds scale.

In both primary and secondary analyses there was limited evidence of heterogeneity in the SNP effect estimates for IVW and MR Egger regression, except for in HDL-C (Q IVW 115.7, p = 0.01; Q MR Egger 115.6, p = 0.01) (**S5 and S6 Tables**).

MR Egger intercepts also indicated limited evidence of directional pleiotropy (**S7 and S8 Tables**). There were no clear outliers in both scatter and leave-one-out plots (**Figs 3 and** S3-S6) and MR-PRESSO detected no individual outliers (**S9 Table**). Where there was evidence of violation of the NOME assumption for the HMGCR, NPC1L1 and CETP instruments (i.e., $I^2$ statistic <0.90) (**S10 Table**), MR-Egger was performed with SIMEX correction and effects were still consistent with the null (**S11 Table**). To further investigate whether the positive findings for *PCSK9* and *LDLR* are due to violation of the exclusion restriction assumption, colocalisation analysis was carried out [49]. This showed no conclusive evidence of shared causal variants between LDL-C and oral/oropharyngeal cancer, with posterior probabilities of 0.055 for *PCSK9* and 0.026 for *LDLR*, respectively (**S12 Table**).

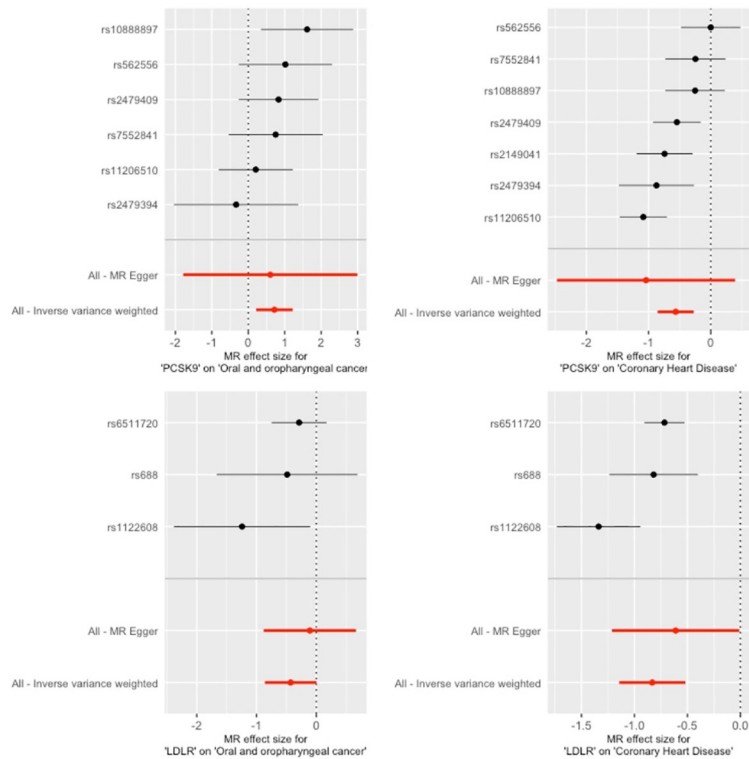

**Fig 2. Forest plots showing the causal effects of cholesterol-lowering PCSK9 and LDLR single nucleotide polymorphisms on coronary heart disease and combined oral/ oropharyngeal cancer in GAME-ON.** Effect estimates on oral and oropharyngeal cancer are reported on the log odds scale.

## Replication in UK Biobank and meta-analysis of results

Primary MR analyses as described above were replicated in UK Biobank, showing limited evidence of an effect of genetically-proxied inhibition of HMGCR, NPC1L1, CETP, PCSK9 and LDLR on risk of OC and OPC (**S13 Table**). Following IVW fixed-effects meta-analysis of GAME-ON and UK Biobank MR results, there was a consistently strong effect of genetically-proxied PCSK9 inhibition on combined OC and OPC (OR IVW 1.8; 95%CI 1.2, 2.8), with good concordance between studies ($I^2$ = 22%) and methods used (**Figs 4 and S7**). Effects for PCSK9 appeared stronger in relation to OPC (OR IVW 2.6; 95%CI 1.4, 4.9) than OC (OR IVW 1.4; 95%CI 0.8, 2.4), but with moderate heterogeneity between studies ($I^2$ = 41%) (**Fig 4**). Conversely, the protective effect for LDLR on OC and OPC combined was also consistent in the meta-analysis (OR IVW 0.7; 95%CI 0.5, 1.0), with good concordance between studies ($I^2$ = 0%) (**Figs 5 and S7**). However, the protective effect seen specifically in relation to OC in GAME-ON (OR IVW 0.5; 95%CI 0.3, 0.9) was not replicated in UK Biobank (OR IVW 1.6; 95%CI 0.5, 4.8), with strong evidence of heterogeneity between the studies ($I^2$ = 66%).

## Discussion

We found limited evidence for a role of cholesterol-lowering in OC and OPC risk. This included the absence of a protective effect of genetically-proxied inhibition of HMGCR, suggesting previous observational studies investigating the relationship between statins and head and neck cancer risk may be subject to residual confounding or bias. However, we did observe an adverse effect of PCSK9 inhibition on OC and OPC risk, which was of a similar magnitude

**Table 3. Mendelian randomization results of circulating lipid traits with risk of oral and oropharyngeal cancer in GAME-ON.**

| Target | N SNPs | Outcome | IVW OR (95% CI) | *P*-value |
|---|---|---|---|---|
| LDL-C | 77 | Oral/ Oropharyngeal cancer | 0.98 (0.87, 1.11) | 0.79 |
| | | Oral cancer | 0.99 (0.85, 1.15) | 0.88 |
| | | Oropharyngeal cancer | 1.03 (0.87, 1.21) | 0.76 |
| HDL-C | 85 | Oral/ Oropharyngeal cancer | 0.98 (0.83, 1.16) | 0.79 |
| | | Oral cancer | 1.08 (0.88, 1.32) | 0.45 |
| | | Oropharyngeal cancer | 0.87 (0.71, 1.05) | 0.15 |
| Total triglycerides | 54 | Oral/ Oropharyngeal cancer | 1.19 (1.01, 1.04) | 0.04 |
| | | Oral cancer | 1.19 (0.96, 1.46) | 0.11 |
| | | Oropharyngeal cancer | 1.19 (0.96, 1.47) | 0.12 |
| Total cholesterol | 82 | Oral/ Oropharyngeal cancer | 1.04 (0.91, 1.18) | 0.55 |
| | | Oral cancer | 1.09 (0.93, 1.28) | 0.30 |
| | | Oropharyngeal cancer | 1.00 (0.84, 1.20) | 0.96 |
| Apolipoprotein A | 9 | Oral/ Oropharyngeal cancer | 0.87 (0.73, 1.03) | 0.11 |
| | | Oral cancer | 0.90 (0.73, 1.12) | 0.34 |
| | | Oropharyngeal cancer | 0.82 (0.65, 1.04) | 0.10 |
| Apolipoprotein B | 14 | Oral/ Oropharyngeal cancer | 1.04 (0.88, 1.22) | 0.67 |
| | | Oral cancer | 1.15 (0.89, 1.49) | 0.29 |
| | | Oropharyngeal cancer | 1.00 (0.83, 1.21) | 0.98 |

Abbreviations: IVW, inverse variance weighted; OR, odds ratio; CI, confidence intervals.

IVW OR represents the exponential change in odds of oral/ oropharyngeal squamous cell carcinoma per SD increase in the circulating lipid trait (one SD for LDL-C = 38.7 mg/dL, HDL-C = 15.5 mg/dL, Apolipoprotein A = 0.32 g/L, Apolipoprotein B = 0.52 g/L, Total triglycerides = 90.7 mg/dL, Total cholesterol = 41.8 mg/dL).

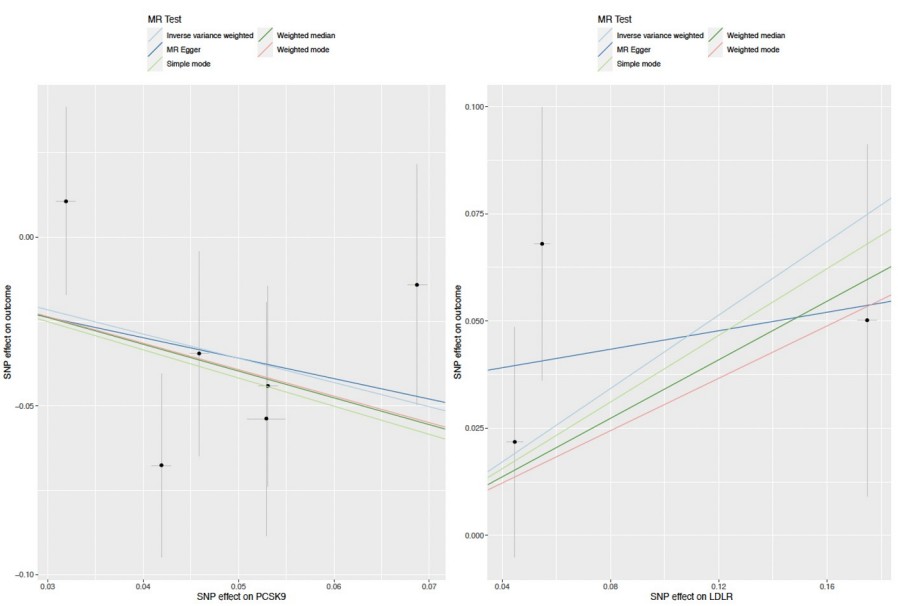

**Fig 3. Scatter plots for LDLR and PCSK9 single nucleotide polymorphisms effect on combined oral/ oropharyngeal cancer in GAME-ON.**

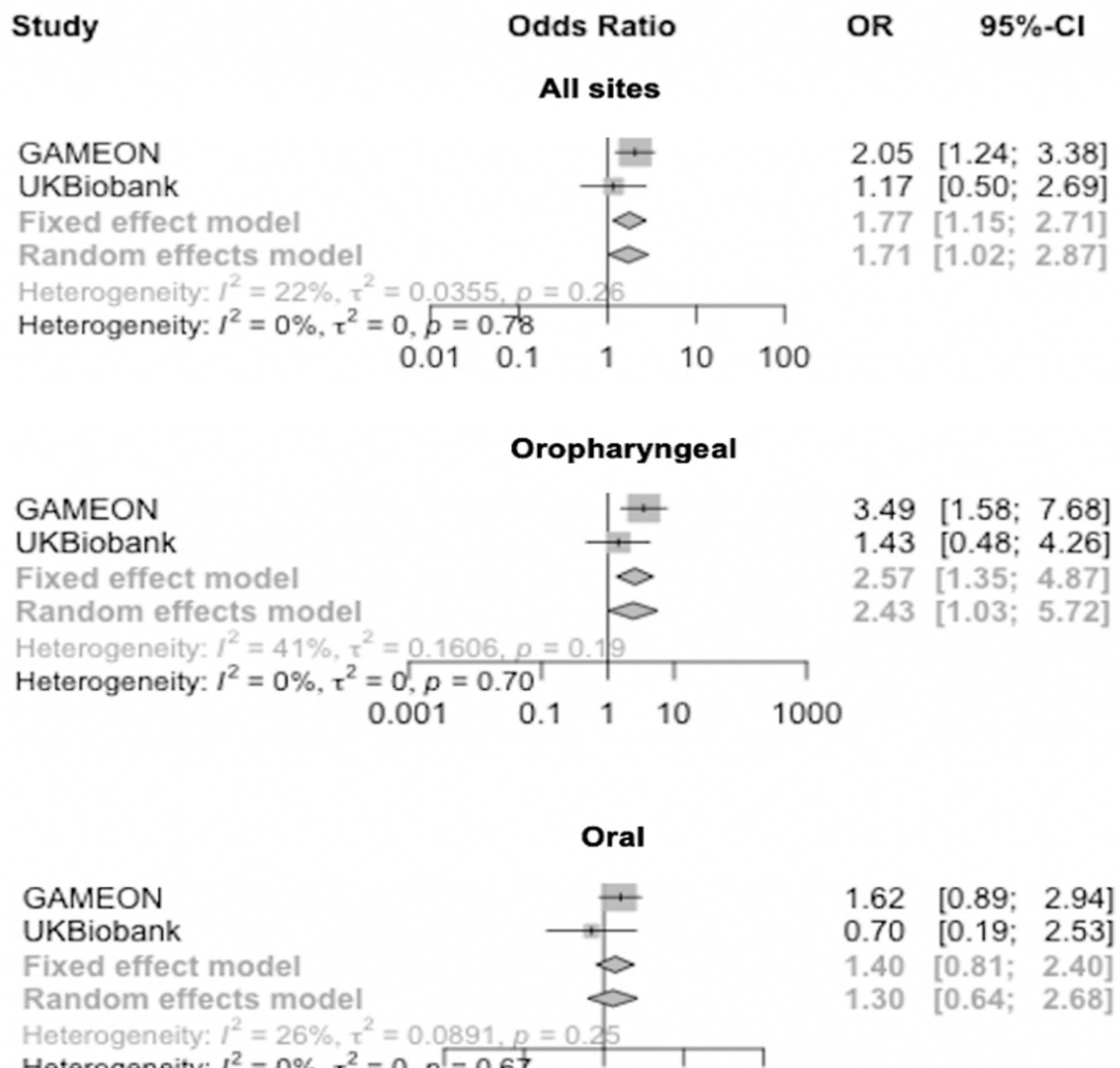

**Fig 4. Forest plots showing inverse variance weighted meta-analysis effects of cholesterol-lowering *PCSK9* single nucleotide polymorphisms on head and neck cancer subsites.**

to the protective effect seen in relation to cardiovascular disease using the same genetic instrument (**Fig 2**). This PCSK9 effect was evident in both the GAME-ON (n = 6,034 OC and OPC cases and n = 6,585 controls) and UK Biobank datasets (n = 839 OC and OPC cases and n = 372,016 controls). We also found some evidence for a protective effect of cholesterol-lowering variants in *LDLR* on OC and OPC risk in both studies. The IVW analysis for HMGCR, NPC1L1, CETP, PCSK9 and LDLR accounting for LD structure followed the same pattern as the main results (**S3 Table**). Further colocalisation analysis for *PCSK9* and *LDLR* showed no conclusive evidence of shared causal variants between LDL-C and oral and oropharyngeal

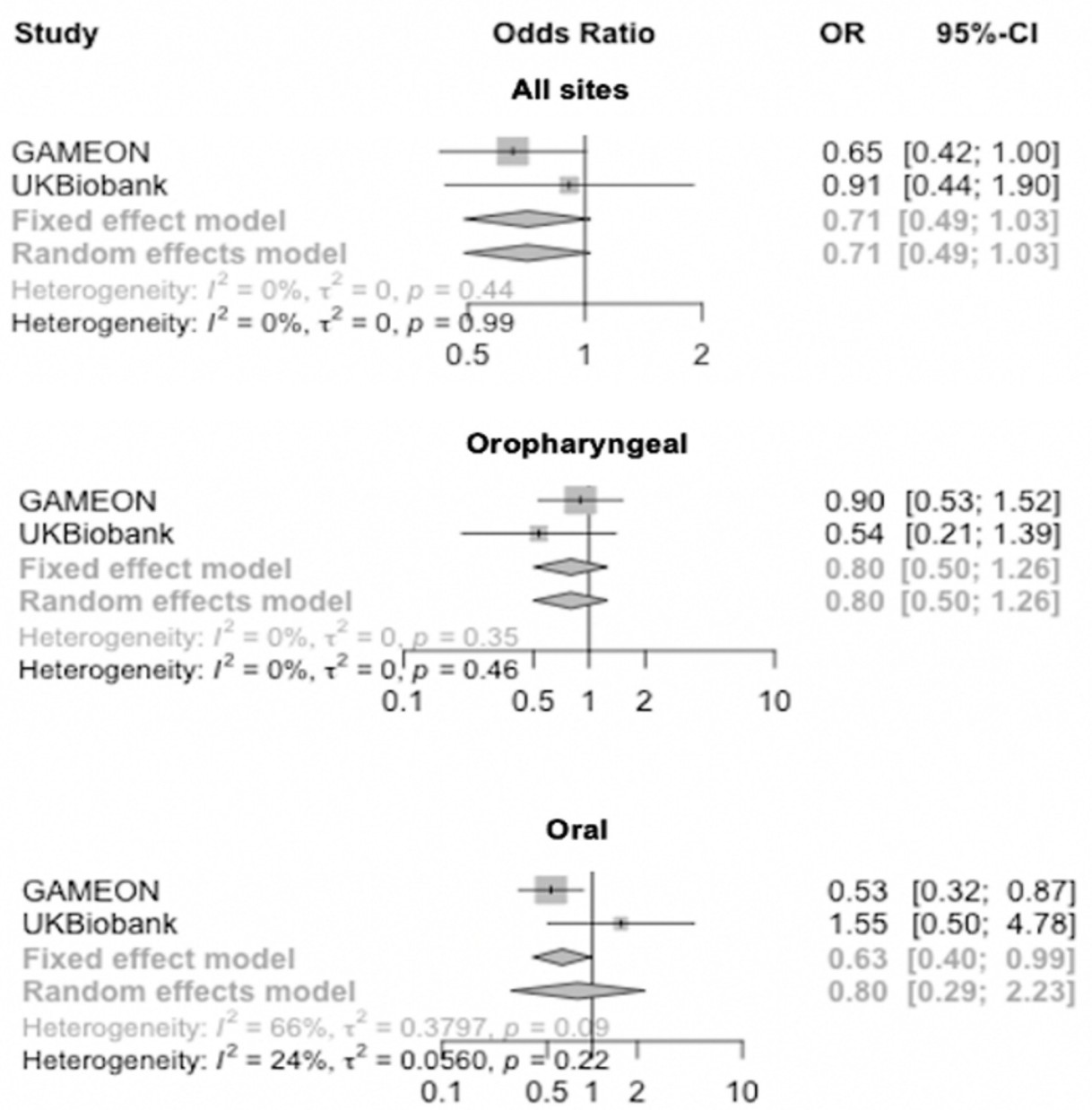

**Fig 5. Forest plots showing inverse variance weighted meta-analysis effects of cholesterol-lowering *LDLR* single nucleotide polymorphisms on head and neck cancer subsites.**

cancer; however, the outcome dataset had a relatively smaller sample size, and typically large sample sizes are required across both traits for better accuracy (**S12 Table**) [49, 55].

### Association between PCSK9, LDLR and cancer risk

Despite the lack of consistent evidence for a role of cholesterol-lowering on risk of OC or OPC in this study, individual effects of both PCSK9 and LDLR were demonstrated which may implicate a role for these drug targets in the development of OC or OPC via other mechanisms.

The effects identified in the present study are directionally consistent with a recent MR analysis of 1,615 combined head and neck cancer cases from UK Biobank, which also found that a 1 SD unit increase in LDL-C proxied by PCSK9 and LDLR was associated with a reduction (OR 0.7 95%CI 0.4, 1.4, p = 0.35) and increase in odds (OR 1.6 95%CI 1.0, 2.4, p = 0.05) of head and neck cancer, respectively [29]. The opposing effects of PCSK9 and LDLR in this study also suggests that cholesterol lowering is unlikely to be the main mechanism of action. However, this previous analysis is limited by the relatively small number of cases and heterogeneity of head and neck cancer subtypes, with no selection for the histological subtype of squamous cell carcinoma. In the present study, we focused specifically on oral and oropharyngeal subtypes of HNSCC.

Beyond the established role of *PCSK9* in cholesterol homeostasis, other potential pleiotropic effects are not well understood. Variants in *PCSK9* have been associated with an increased risk of diabetes (OR 1.1, 95%CI 1.0, 1.2 for each 10 mg per decilitre decrease in LDL-C) [28]. However, a recent phenome-wide association study (PheWAS) did not find *PCSK9* or *LDLR* to be correlated with any non-lipid-related phenotypes, including diabetes [56]. There is limited *in vivo* and *in vitro* evidence that *PCSK9* might be involved in both cell proliferation and apoptosis. The gene was initially designated as *NARC1* (neural apoptosis-regulated convertase 1), involved in apoptosis of cerebellar neurons [57] and *PCSK9* has since been found to be upregulated in some cancers [58, 59].

One suggested mechanism for a link with cancer progression is that the increased expression of *PCSK9* prevents LDL-receptor (LDLR) recycling, leading to hypercholesterolaemia and more exogenous lipid to support the proliferation of the tumour [57]. Our study suggests the opposite, that genetically-proxied inhibition of *PCSK9* results in an increased risk of OC or OPC. We hypothesise that access to intracellular LDL-C could in fact be pro-tumourigenic, providing a favourable environment for a developing tumour cell, maintaining membrane integrity and promoting cell division. Therapies such as statins, ezetimibe, and PCSK9 inhibitors may all lower LDL-C level through the upregulation of LDL-receptors, resulting in elevated intracellular cholesterol. However, cancer mechanisms are often context dependent and perhaps only the expression of *PCSK9* and *LDLR* is relevant in head and neck cancer. CETP inhibitors instead block the transfer of cholesteryl ester from HDL-C to LDL-C, thereby raising HDL-C and lowering LDL-C (and apolipoprotein B), as well as enhancing reverse cholesterol transport [60]. Unlike statins, CETP inhibitors do not appear to increase the risk of type 2 diabetes, thought to be as a result of pancreatic islet cell cholesterol accumulation with use of other cholesterol-lowering drugs [61]. Therefore, the absence of effect when proxying *CETP* inhibition in this study, supports the possible mechanism of LDL-C uptake via the LDL-receptor in OC or OPC.

While both *PCSK9* [62] and *LDLR* [62] are expressed in head and neck tumours, this was not evident in normal oral or oropharyngeal tissue, and sufficient tissue is currently not available in expression datasets [63]. Recent studies have associated elevated *PCSK9* with alcohol use disorder, including the interesting possibility of using anti-PSCK9 monoclonal antibodies for the treatment of alcoholic liver disease [64, 65]. Given that alcohol is a well-known risk factor for HNSCC, this pleiotropy could have partially explained the effect seen in our study, however we proxied the inhibition of *PCSK9* and so would have expected to see a protective effect of this gene in HNSCC cases who may have been heavy alcohol drinkers. Further investigation is required to untangle the relationship between *PCSK9*, alcohol and head and neck cancer, including a subsequent multivariable MR analysis.

## Comparison with previous studies

It is believed that statins could play a potential role in cancer chemoprevention which may reduce the risk of some site-specific cancers such as prostate [66] and ovarian [30], but not all. Some of these studies have reported cancer risk reductions by as much as 50–65% [67–69]. However, meta-analyses and clinical trials have contradicted these findings [70, 71]. In addition to confounding, immortal time bias may have inflated observational results because, to be classified as a long-term statin user necessitates that users survived without cancer over a long period [19, 72]. Dickerman et al. used electronic records from 733,804 adults with 10-year follow-up to emulate a trial design. To achieve this a pre-specified protocol was set, including eligibility criteria and checks were made to ensure effect estimates for statins on cancer were comparable between the large observational dataset and trial. The authors found little indication that statin therapy influences cancer incidence, which was consistent with the analyses of randomised trials (with a 10-year cancer-free survival difference of −0.3% 95%CI −1.5%, 0.5%) [19]. Nonetheless, recent MR studies [29, 30] have identified an association between variants in *HMGCR* with cancer risk, but not alternative cholesterol-lowering treatments or genetically-predicted LDL-C, suggesting that statins may reduce cancer risk through a cholesterol independent pathway. A recent case-control study of over 11,000 participants found an inverse association between statin use and the occurrence of HNSCC (OR 0.86, 95%CI 0.77, 0.95, p = <0.01, of prior statin exposure for cases compared to propensity score-matched controls) [18]. However, this observational study failed to stratify by anatomical subsite, which may have revealed distinct associations given differences in aetiology (e.g., the strong association of HPV infection with OPC). Furthermore, a wider systematic review found that the evidence for the role of statins in the prevention of HNSCC was limited [73]. As HNSCC incidence is a rare outcome, randomised controls trials are not feasible, so we must be cautious interpreting the available observational findings given the potential for bias and confounding as discussed previously.

In contrast to previous MR studies assessing overall cancer risk [29] and ovarian specific risk [30], the MR carried out here in relation to OC and OPC showed no effect using genetic instruments for HMGCR (statins). There was also limited evidence for a causal effect of NPC1L1 (ezetimibe), CETP (CETP inhibitors) as well as a number of other circulating lipid traits on OC or OPC. Therefore, it remains unclear as to whether the effects observed with PCSK9 and LDLR are via LDL-C lowering or another less well-established pathway [74, 75], such as receptor regulation for viral entry, synthesis of sex hormones and resultant dysregulated metabolism [76–78].

## Strengths and limitations of this study

Protective associations between cholesterol-lowering therapies such as statins and head and neck cancer risk seen in previous observational studies could be a result of reverse causation, immortal time bias, lack of randomisation or confounding by socioeconomic status, smoking or HPV infection, for example. Our study applied MR in an attempt to overcome these issues, using the largest number of SNPs identified from the latest GWAS for both cholesterol-lowering and head and neck cancer that could be identified in the literature [25, 33]. A series of pleiotropy-robust MR methods and outlier detection were also applied to rigorously explore the possibility that findings were not biased as a result of pleiotropy. However, there was no HPV data available in these summary results to enable more detailed stratified analysis of OPC. We did replicate findings for PCSK9 and LDLR, but the number of cases are low. Finally, most participants in the GAME-ON network [33] were of European or North American decent, with only around 11% from South America, and participants included in the UK Biobank

analysis were exclusively of European descent, so more work is required to determine if our results translate to other ancestry groups.

In conclusion, our MR analyses provided little evidence for a role of cholesterol lowering in OC or OPC risk although effects of genetically-proxied inhibition of *PCSK9* and cholesterol-lowering variants in *LDLR* were observed in relation to OPC and OC risk. Given the lack of a common pathway to carcinogenesis in OC or OPC, identifying metabolic targets that may be common to all tumours, regardless of the activated molecular pathway, could help simplify a preventative or therapeutic approach [79]. Replication of our findings in other head and neck cancer datasets and use of individual-level follow-up data with HPV status could provide further insight into the effect of these genetic instruments on risk, treatment outcomes and survival in head and neck cancer.

## Supporting information

**S1 STROBE checklist. A checklist of items that should be included in reports of observational studies.**
(DOC)

**S1 Table. Detailed summary of genetic variants proxying circulating LDL-C, HDL-C, total triglyceride, total cholesterol, apolipoprotein A and B levels.** EA, effect allele or low-density lipoprotein-cholesterol (LDL-C) lowering allele; OA, other or non-effect allele; EAF, effect allele frequency; se, standard error. Beta represents the change in LDL-C levels per copy of the effect allele.
(DOCX)

**S2 Table. Genetic correlation results for HMGCR, NPC1L1, CETP, PCSK9 and LDLR single nucleotide polymorphisms.**
(DOCX)

**S3 Table. Mendelian randomization results of genetically-proxied inhibition of HMGCR, NPC1L1, CETP, PCSK9 and LDLR with risk of combined oral/ oropharyngeal cancer accounting for LD structure in GAME-ON.**
(DOCX)

**S4 Table. Assessing weak instrument bias (F-statistic) and proportion of variance in the phenotype ($R^2$) explained by the genetic instruments.**
(DOCX)

**S5 Table. Assessing heterogeneity of single nucleotide polymorphism effect estimates in inverse-variance weighted (IVW) and MR Egger regression for primary analysis.** Abbreviations: Q, Q-statistic; df, degrees of freedom; P, p-value.
(DOCX)

**S6 Table. Assessing heterogeneity of single nucleotide polymorphism effect estimates in inverse-variance weighted (IVW) and MR Egger regression for secondary analysis.** Abbreviations: Q, Q-statistic; df, degrees of freedom; P, p-value.
(DOCX)

**S7 Table. Assessing directional pleiotropy through MR Egger intercept for primary analysis.** Abbreviations: UVMR, univariable Mendelian randomization; SE, standard error; P, p-value.
(DOCX)

**S8 Table. Assessing directional pleiotropy through MR Egger intercept for secondary analysis.** Abbreviations: UVMR, univariable Mendelian randomization; SE, standard error; P, p-value.
(DOCX)

**S9 Table. MR-PRESSO results for HMGCR, NPC1L1, CETP, PCSK9, LDLR and other lipid trait SNPs on combined oral/ oropharyngeal cancer.** Abbreviations: RSSobs, residual sum of squares observations.
(DOCX)

**S10 Table. Assessing violation of the "NO Measurement Error" (NOME) assumption for instruments used in MR-Egger regression.** Abbreviations: $I^2$, I-squared statistic.
(DOCX)

**S11 Table. SIMEX correction MR Egger regression results for HMGCR, NPC1L1 and CETP (where $I^2$ <0.90).** Abbreviations: OR, odds ratio; CIL, lower confidence interval; CIU, upper confidence interval; P, p-value.
(DOCX)

**S12 Table. Colocalisation results for PCSK9 and LDLR with oral and oropharyngeal cancer combined and LDL-C.**
(DOCX)

**S13 Table. Mendelian randomization results of genetically-proxied inhibition of HMGCR, NPC1L1, CETP, PCSK9 and LDLR with risk of oral and oropharyngeal cancer including sensitivity analyses in UK Biobank.** Abbreviations: IVW, inverse variance weighted; OR, odds ratio; CI, confidence intervals; P, p-value. OR represents the exponential change in odds of oral/ oropharyngeal squamous cell carcinoma per genetically-proxied inhibition of drug target equivalent to a 1 mmol/L decrease in LDL-C.
(DOCX)

**S1 Fig. Forest plots showing Mendelian randomization results for genetically-proxied inhibition of HMGCR, NPC1L1, CETP, PCSK9 and LDLR with risk of combined oral/ oropharyngeal cancer in GAME-ON.** Effect estimates on oral and oropharyngeal cancer are reported on the log odds scale.
(PDF)

**S2 Fig. Forest plots showing Mendelian randomization results for LDL-C, HDL-C, total cholesterol, total triglycerides, Apolipoprotein A and Apoprotein B single nucleotide polymorphisms effect on combined oral/oropharyngeal cancer in GAME-ON.** Effect estimates on oral and oropharyngeal cancer are reported on the log odds scale.
(PDF)

**S3 Fig. Scatter plots for HMGCR, NPC1L1 and CETP single nucleotide polymorphisms effect on combined oral/oropharyngeal cancer in GAME-ON.**
(PDF)

**S4 Fig. Leave one out analysis for HMGCR, NPC1L1, CETP, PCSK9 and LDLR single nucleotide polymorphisms effect on combined oral/ oropharyngeal cancer in GAME-ON.**
(PDF)

**S5 Fig. Scatter plots for LDL-C, HDL-C, total cholesterol, total triglycerides, apolipoprotein A and apolipoprotein B single nucleotide polymorphisms effect on combined oral/**

**oropharyngeal cancer in GAME-ON.**
(PDF)

**S6 Fig. Leave one out analysis for LDL-C, HDL-C, total cholesterol, total triglycerides, Apolipoprotein A and Apoprotein B single nucleotide polymorphisms effect on combined oral/oropharyngeal cancer in GAME-ON.**
(PDF)

**S7 Fig. Forest plots showing meta-analysed causal effects of cholesterol-lowering HMGCR, NPC1L1, CETP, LDLR and PCSK9 single nucleotide polymorphisms on combined head and neck cancer in UK Biobank and GAME-ON.**
(PDF)

## Author Contributions

**Conceptualization:** Mark Gormley, Emma E. Vincent, Rebecca C. Richmond.

**Data curation:** Mark Gormley.

**Formal analysis:** Mark Gormley, Kimberley Burrows, Rebecca C. Richmond.

**Funding acquisition:** Mark Gormley.

**Investigation:** Mark Gormley.

**Methodology:** Mark Gormley, James Yarmolinsky, Tom Dudding, Kimberley Burrows, Richard M. Martin, Jessica Tyrrell, Brenda Diergaarde, Andrew R. Ness, George Davey Smith, Rebecca C. Richmond.

**Project administration:** Mark Gormley.

**Resources:** Steven Thomas, Miranda Pring, Stefania Boccia, Andrew F. Olshan, Brenda Diergaarde, Rayjean J. Hung, Geoffrey Liu, Eloiza H. Tajara, Patricia Severino, Martin Lacko.

**Supervision:** Jessica Tyrrell, Andrew R. Ness, George Davey Smith, Emma E. Vincent, Rebecca C. Richmond.

**Validation:** Mark Gormley, Rebecca C. Richmond.

**Visualization:** Mark Gormley, Rebecca C. Richmond.

**Writing – original draft:** Mark Gormley, James Yarmolinsky, Tom Dudding, Kimberley Burrows, Richard M. Martin, Miranda Pring, Brenda Diergaarde, Andrew R. Ness, George Davey Smith, Emma E. Vincent, Rebecca C. Richmond.

**Writing – review & editing:** Mark Gormley, James Yarmolinsky, Tom Dudding, Kimberley Burrows, Richard M. Martin, Steven Thomas, Jessica Tyrrell, Paul Brennan, Miranda Pring, Stefania Boccia, Andrew F. Olshan, Brenda Diergaarde, Rayjean J. Hung, Geoffrey Liu, Danny Legge, Eloiza H. Tajara, Patricia Severino, Martin Lacko, Andrew R. Ness, George Davey Smith, Emma E. Vincent, Rebecca C. Richmond.

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
