## [Decision Letter · Decision Letter 0]

4 Jan 2021

Dear Dr Gormley,

Thank you very much for submitting your Research Article entitled 'Using genetic variants to evaluate the causal effect of cholesterol lowering on head and neck cancer risk: a Mendelian randomization study' to PLOS Genetics.

The manuscript was fully evaluated at the editorial level and by two independent peer reviewers. The reviewers appreciated the attention to an important problem, but raised some concerns about the current manuscript. Based on the reviews, we will not be able to accept this version of the manuscript, but we will be pleased to review a revised version. We cannot, of course, promise publication at that time.

If you decide to revise the manuscript for further consideration at PLOS Genetics, please aim to resubmit within the next 60 days, unless it will take extra time to address the concerns of the reviewers, in which case we would appreciate an expected resubmission date by email to plosgenetics@plos.org.

[LINK]

Please do not hesitate to contact us if you have any concerns or questions.

Yours sincerely,

Thilo Dörk, Ph.D.

Guest Editor

PLOS Genetics

David Kwiatkowski

Section Editor: Cancer Genetics

PLOS Genetics

Reviewer's Responses to Questions

**Comments to the Authors:**

Reviewer #1: This is very well written and explained manuscript. The study is well designed, with the relationship between cholesterol and head and neck cancer comprehensively studied. The analyses are appropriate and the conclusions drawn sound. I only have minor comments for consideration.

Introduction: Were the observational studies assessing head and neck cancer risk and survival performed stratifying by OC and OPC? Given the differences in etiology for these cancers (e.g. HPV exposure) some discussion of this could be worthwhile.

While I appreciate that the authors haven’t overstated their findings, some mention of multiple testing should be made in the manuscript.

Reviewer #2: This is an interesting manuscript investigating the association of cholesterol lowering in relation to head and neck cancer risk using Mendelian randomization. The research is well-conducted. I have a few comments/suggestions which may further improve this work.

1) Given that head and neck cancer is a relatively rare cancer in the western world and that the number of cases was 6K in the current study, I would have liked to see a power analysis to check the minimum detectable OR and compare to findings observed in observational studies.

2) The authors were very detailed in the sensitivity analyses they performed to probe into potential violations of MR assumptions. One analysis that was missing was a co-localisation analysis that can investigate whether the positive findings for PCSK9 and LDLR are due to violation of the exclusion restriction assumption due to LD.

3) In the Discussion, the authors nicely discuss the potential association between PCSK9-alcohol and head & neck cancer. A multivariable MR analysis adjusting for alcohol is warranted, although power may be low.

4) The authors made a comment in lines 553-555 about potential for immortal time bias in observational studies, but this bias is far more prevalent in studies of cancer survival than incidence.

5) The last sentence of the Conclusion seems unsubstantiated based on the findings of the current literature and the current paper.

**Have all data underlying the figures and results presented in the manuscript been provided?**

Reviewer #1: Yes

Reviewer #2: Yes

PLOS authors have the option to publish the peer review history of their article (what does this mean?). If published, this will include your full peer review and any attached files.

Reviewer #1: No

Reviewer #2: No

---

## [Editor Report · Decision Letter 1]

31 Mar 2021

Dear Dr Gormley,

We are pleased to inform you that your manuscript entitled "Using genetic variants to evaluate the causal effect of cholesterol lowering on head and neck cancer risk: a Mendelian randomization study" has been editorially accepted for publication in PLOS Genetics. Congratulations!

Yours sincerely,

Thilo Dörk, Ph.D.

Guest Editor

PLOS Genetics

David Kwiatkowski

Section Editor: Cancer Genetics

PLOS Genetics

Comments from the reviewers (if applicable):

**Data Deposition**

http://datadryad.org/submit?journalID=pgenetics&manu=PGENETICS-D-20-01584R1

**Press Queries**

---

## [Editor Report · Acceptance letter]

14 Apr 2021

PGENETICS-D-20-01584R1 

Using genetic variants to evaluate the causal effect of cholesterol lowering on head and neck cancer risk: a Mendelian randomization study 

Dear Dr Gormley, 

We are pleased to inform you that your manuscript entitled "Using genetic variants to evaluate the causal effect of cholesterol lowering on head and neck cancer risk: a Mendelian randomization study" has been formally accepted for publication in PLOS Genetics! Your manuscript is now with our production department and you will be notified of the publication date in due course.

With kind regards,

Katalin Szabo

PLOS Genetics

On behalf of:
